# WAKENING PAST CONCEPTS WITHOUT PAST DATA: CLASS-INCREMENTAL LEARNING FROM PLACEBOS

## ABSTRACT

Not forgetting knowledge about previous classes is one of the key challenges in class-incremental learning (CIL). A common technique to address this challenge is knowledge distillation (KD) that penalizes inconsistencies across models of subsequent phases. As old-class data is scarce, the KD loss mainly uses new class data. However, we empirically observe that this both harms learning of new classes and also underperforms to distil old class knowledge from the previous phase model. To address this issue, we propose to compute the KD loss using placebo data chosen from a free image stream (e.g., Google Images), which is both simple and surprisingly effective even when there is no class overlap between the placebos and the old data. When the image stream is available, we use an evaluation function to quickly judge the quality of candidate images (good or bad placebos) and collect good ones. For training this function, we sample pseudo CIL tasks from the data in the $0$-th phase and design a reinforcement learning algorithm. Our method does not require any additional supervision or memory budget, and can significantly improve a number of top-performing CIL methods, in particular on higher-resolution benchmarks, e.g., ImageNet-1k and ImageNet-Subset, and with a lower memory budget for old class exemplars, e.g., five exemplars per class.

## 1 INTRODUCTION

AI learning systems are expected to learn new concepts while maintaining the ability to recognize old ones. In many practical scenarios, they cannot access the past data due to the limitations such as storage or data privacy but are required to recognize all seen classes. Motivated by this, Rebuffi et al. (2017) formulated the so-called class-incremental learning (CIL) problem, where training data of different classes gradually comes phase-by-phase, and the model keeps on re-training on new class data and is evaluated on both new and old classes. It is obvious that re-training in each new phase tends to override knowledge acquired from old data (McCloskey & Cohen, 1989; McRae & Hetherington, 1993; Ratcliff, 1990), which is usually called "catastrophic forgetting". To alleviate this issue, most class-incremental learning methods (Rebuffi et al., 2017; Li & Hoiem, 2016; Hou et al., 2019; Douillard et al., 2020; Liu et al., 2020a; 2021a) are equipped with a knowledge distillation (KD) loss that penalizes any feature or prediction inconsistency between the models in adjacent phases. The ideal KD loss in each phase should be computed on the original old class data, which, however, is impossible in the constrained settings of CIL. Existing methods use new class data as the substitute (to compute the KD loss). We argue that this 1) hampers the learning of new classes as it distracts the model from fitting the ground truth labels of new classes, and 2) falls short of distilling the knowledge of old classes since using new class data can not generate the same soft labels (or features) as using old data. Quantitative results are shown in Figure 1 where "old" denotes the upper bound (when using old data) and "new" shows rather poor results when using new data for KD.

In this paper, instead, we propose to search placebo images online to compute the KD loss for old classes. Ideally, the distillation effect of using our placebo images is close to that of using original old data. To optimize this effect, we propose a reinforcement learning (RL) algorithm where we define the state, action and policy function under the constraints of the CIL setting. The aim of the policy is to produce phase-specific evaluation functions (for evaluating the quality of placebo images) that are able to handle the dynamic states along with incremental phases, e.g., the function

---

We made the corresponding changes in the revised paper and colorized these changes in deep blue.

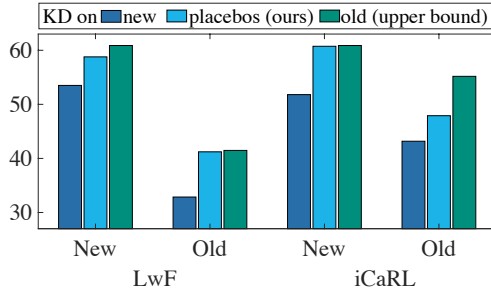
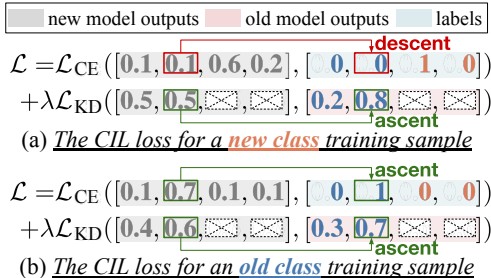

Figure 1: **Left**: Average accuracy (%) of new and old classes in the last phase, respectively using LwF (Li & Hoiem, 2016) and iCaRL (Rebuffi et al., 2017). The **dark blue**, **light blue**, and **green** bars indicate the results of using new data, placebos, and old data to compute KD losses, respectively. "KD on new" means computing the KD loss on both new data and old exemplars. All hyperparameters follow Hou et al. (2019). **Right**: Conceptual illustrations of the loss functions applied to new and old samples. The **dark blue** and **orange** numbers denote the predictions of old and new classes, respectively. It is clear in (a) that the objectives are different when using a new class sample to do KD (the oracle case is to have both "ascent"), e.g., the ground truth label for the second old class is 0, while the "KD label" at this position is 0.8. This is not an issue when using an old class sample, e.g., in (b), its ground truth label and "KD label" have consistent magnitudes at the same position (1 and 0.7, respectively). Please note the KD loss is softmax KL divergence loss.

in a later phase is expected to handle more complex evaluations on the placebos of more old classes. The intuitive objective of the policy is to maximize the cumulative accuracy across all incremental phases. However, this does not comply with the CIL setting, where neither past nor future data are accessible. To overcome this problem, we propose to pre-train the policy on pseudo CIL tasks which can be built on any available categorical data, e.g., the data in the 0-th When RL is done (e.g., in the 0-th phase), the policy function will be deployed in each incremental phase. Specifically, we first leverage this policy to generate an evaluation function according to the state of the phase and then use the function to estimate the quality of each candidate image (a good or bad placebo) in the batch loaded from a free and unlabeled data stream. We keep only high-quality placebos to compute the KD loss, after which we discard all of them and load another batch (and repeat the above steps).

We highlight that our method does not require any additional supervision or annotation. Storing the free data batch needs only a tiny amount of additional memory, which we can obtain by deleting a few new class samples without harming the performance. This memory is reused for every batch, as the batch data are immediately discarded after computing the KD loss. Figure 1 shows that our method (light blue) indeed alleviates the KD issue caused by using new class data (dark blue). It is encouraging that our method achieves the upper bound performance for recognizing the old (new) classes when plugged into LwF (Li & Hoiem, 2016) (iCaRL (Rebuffi et al., 2017)). In the experiments below, we evaluate it more extensively by incorporating it into multiple strong baselines such as PODNet (Douillard et al., 2020), LUCIR (Hou et al., 2019) and AANets (Liu et al., 2021a), and inspect its performance carefully by conducting a series of ablative studies. Our results on three popular CIL benchmarks show clear and consistent improvements over the baselines, especially on higher-resolution image datasets (such as ImageNet-Subset and ImageNet-1k) and with lower memory budgets for old class exemplars (e.g., five exemplars per class).

Our technical contribution is three-fold. 1) A generic CIL method that leverages unlabeled placebo data from a free image stream to improve the KD effect for both logit distillation and feature distillation methods. 2) A reinforcement-learning-based algorithm that learns a policy to adaptively produce phase-specific functions to evaluate the quality of placebos. 3) Extensive comparisons and visualizations for our method in three CIL benchmarks, taking top-performing models as baselines.

## 2 RELATED WORK

**Incremental learning** (also called "**continual learning**") trains a model using the data coming in a sequence of learning phases. Recent methods are focused on either task-incremental learning where in each phase a new task (dataset) comes that contains a new split across all classes, i.e., classes are all seen before (Chaudhry et al., 2019; Davidson & Mozer, 2020; Hu et al., 2019; Riemer et al., 2019;

Shin et al., 2017; Zhao et al., 2020), or class-incremental learning (CIL) where each phase contains the data of a set of new classes, i.e., classes are unseen before. This paper is focused on CIL. The key challenge in CIL is the forgetting problem—older classes are forgotten in later phases. There are three main lines of work. ***Memory-based*** methods (Rebuffi et al., 2017; Shin et al., 2017) preserve a small number of old class data (called exemplars) and replay the model on them together with new class data. ***Regularization-based*** methods introduce regularization terms in the loss function to consolidate previous knowledge when training the model on new data. The key idea is to enforce model prediction logits (Li & Hoiem, 2016; Rebuffi et al., 2017), feature maps (Douillard et al., 2020; Hou et al., 2019), the topology in the feature space (Tao et al., 2020), or low-dimensional manifolds (Simon et al., 2021) of the new-phase model to be close to that in the pre-phase model. ***Network-architecture-based*** methods (Rusu et al., 2016; Xu & Zhu, 2018; Abati et al., 2020; Liu et al., 2021a) design "incremental network architectures" by expanding the network capacity for new data or freezing partial network parameters to keep the knowledge of old classes. For example, Yan et al. (2021) froze the previously learned representation and augmented it with additional feature dimensions from a new learnable feature extractor. Our method belongs to the ***Regularization-based*** group. Some prior papers use unlabeled data to improve regularization. Lee et al. (2019) proposed a confidence-based sampling method to leverage unlabeled external data for their global distillation loss. Zhang et al. (2020) trained a separate model only for the new classes and consolidated both old and new models by exploiting unlabeled auxiliary data. Liu et al. (2020b) integrated an ensemble of auxiliary classifiers to estimate regularization constraints and used unlabeled data to maximize the classifier discrepancy. Our method differs in two aspects: 1) we design an RL-based algorithm that learns a policy to adaptively produce phase-specific functions to evaluate the quality of the unlabeled data; 2) our method is more generic and can be easily plugged in both logit distillation and feature distillation methods to boost the performance.

**Reinforcement learning** aims to learn how to let intelligent agents take actions in an environment to maximize a cumulative reward. Some incremental learning papers also deploy reinforcement learning algorithms in their frameworks. Xu & Zhu (2018) used RL to adaptively expand its backbone network when a new task arrives, i.e., use RL to determine how many and where to add convolution filters. Liu et al. (2021b) used RL to learn a policy to adjust the memory allocation between old and new class data dynamically along with the learning phases. In this work, they also proposed to create pseudo tasks using the dataset observed in the 0-th phase or transferred from another dataset. Our method differs in two aspects: 1) our RL algorithm is designed for data selection rather than memory allocation; 2) RL is not mandatory in our method. Using heuristic distances such as cosine distance also works for our data section, and incorporating RL can boost the model performance.

## 3 PRELIMINARIES

**Class-incremental learning (CIL)** usually assumes $(N+1)$ learning phases: an initial phase and $N$ incremental phases during which the number of classes gradually increases till the maximum (Douillard et al., 2020; Hou et al., 2019; Hu et al., 2021; Liu et al., 2020a). In the initial (0-th) phase, data $\mathcal{D}_{1:c_0} = \{\mathcal{D}_1, ..., \mathcal{D}_{c_0}\}$, containing the training samples of $c_0$ classes, are used to learn the initial classification model $\Theta_0$. After this phase, only a small subset of $\mathcal{D}_{1:c_0}$ (i.e., exemplars denoted as $\mathcal{E}_{1:c_0} = \{\mathcal{E}_1, ..., \mathcal{E}_{c_0}\}$) can be stored in the memory and used as replay samples in later phases. We use $c_i$ to denote the number of classes we have observed from the 0-th phase to the $i$-th phase. In the $i$-th incremental phase, we get new class data $\mathcal{D}_{c_{i-1}+1:c_i} = \{\mathcal{D}_{c_{i-1}+1}, ..., \mathcal{D}_{c_i}\}$ of $(c_i - c_{i-1})$ classes and load exemplars $\mathcal{E}_{1:c_{i-1}}$ from the memory. Then, we initialize $\Theta_i$ with $\Theta_{i-1}$, and train it using $\mathcal{E}_{1:c_{i-1}} \cup \mathcal{D}_{c_{i-1}+1:c_i}$. The resulting model $\Theta_i$ will be evaluated with a test set for $\mathcal{D}_{1:c_i}$.

**Reinforcement learning (RL)** aims to learn an optimal policy $\pi_\phi$ (parameterized by $\phi$) for an agent interacting in an unknown environment (Williams, 1992; Xu & Zhu, 2018; Zoph & Le, 2017). In the CIL scenario, the agent observes the current state $s_i$ from the environment in each incremental phase, and then takes an action $\boldsymbol{a}_i$ for this phase according to the policy $\pi_\phi(\boldsymbol{a}_i|s_i)$. Subsequently, the environment is updated to a new state $s_{i+1}$ for the next phase and the reward $r_i$ is calculated to optimize the parameters of $\pi_\phi(\boldsymbol{a}_i|s_i)$. Specifically, the learning objective of $\pi_\phi(\boldsymbol{a}_i|s_i)$ is to maximize the expected cumulative reward (i.e., the cumulative validation accuracy of all training CIL tasks) $J(\phi) = \mathbb{E}_{\pi_\phi}[R] = \mathbb{E}_{\pi_\phi}[\sum_{i=0}^{N} r_i]$. We use the reward because the target of the CIL system is at any time providing a competitive classifier for the classes observed so far (Rebuffi et al., 2017).

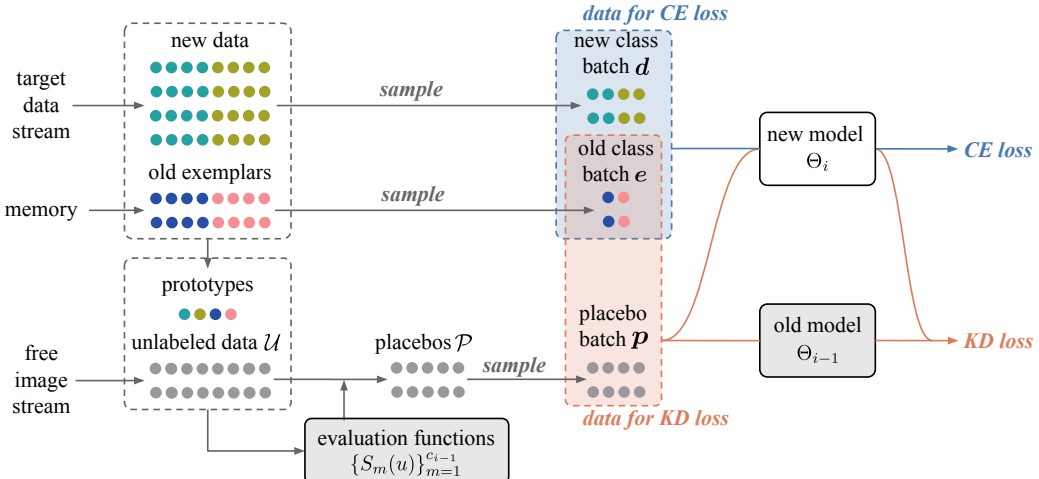

Figure 2: The computation flow of our method PlaceboCIL in the $i$-th phase. We have three kinds of input data: old class exemplars, new class data, and a free stream of unlabeled images. We compute two losses: CE loss on $e \cup d$ and KD loss on $e \cup p$. The way to construct placebos is as follows: 1) We use the evaluation functions $\{S_m(u)\}_{m=1}^{c_{i-1}}$ to output the score for each unlabeled sample. 2) We load a batch of unlabeled data $\mathcal{U}$ from the free image stream, and add $K$ placebos with the highest scores for each old class to $\mathcal{P}$. 3) We compute KD loss using the placebos then delete them from $\mathcal{P}$. 4) When we use up the selected placebos, we repeat the selection steps.

## 4 CLASS-INCREMENTAL LEARNING WITH PLACEBOS (PLACEBOCIL)

In Figure 2, we illustrate the computation flow of our method PlaceboCIL in each incremental phase. We have three kinds of data: old class exemplars stored in the memory, new class data coming in the current phase, and a free stream of unlabeled images. To update the new model, we compute two losses: cross-entropy (CE) loss on the sampled batch of old exemplars and new class data; and KD loss on the sampled exemplars as well as the selected placebos. We elaborate the update process in Section 4.1. As shown in the lower branch of Figure 2, the evaluation is based on the prototypes of all classes observed so far, and the evaluation functions are generated by the policy pre-trained by RL (or heuristic distances without RL). We introduce our RL algorithm in Section 4.2.

### 4.1 TRAINING CIL MODELS WITH PLACEBOS

In the following, we introduce how to evaluate the unlabeled image to get high-quality placebos, and how to use placebos to compute the KD loss in CIL. We summarize the overall training process of an incremental phase in Algorithm 1.

**Building evaluation functions.** We argue that high-quality placebos for the $m$-th old class should meet two requirements: (1) the placebos should be close to the prototype of the $m$-th class in the feature space because they will be used to activate the related neurons of the $m$-th old class in the model; and (2) the placebos should be far from the prototypes of other classes in feature space so that they will not cause the KD issues shown in Figure 1. To achieve these, we design the following evaluation function $\mathcal{S}_m(x)$ for the $m$-th old class in the $i$-th phase:

$$
\mathcal{S}_m(x) = -\underbrace{\cos\left(\mathcal{F}_{\Theta_i}(x), \frac{1}{|\mathcal{E}_m|}\sum_{z\in\mathcal{E}_m}\mathcal{F}_{\Theta_i}(z)\right)}_{\text{ProtoDist}(x,\mathcal{E}_m)} + \beta_i \underbrace{\frac{1}{c_{i-1}-1}\sum_{\substack{n=1\\n\neq m}}^{c_{i-1}}\cos\left(\mathcal{F}_{\Theta_i}(x), \frac{1}{|\mathcal{E}_n|}\sum_{z\in\mathcal{E}_n}\mathcal{F}_{\Theta_i}(z)\right)}_{\text{ProtoDist}(x,\mathcal{E}_{-m})}
$$
$$
+ \gamma_i \underbrace{\frac{1}{c_i-c_{i-1}}\sum_{l=c_{i-1}+1}^{c_i}\cos\left(\mathcal{F}_{\Theta_i}(x), \frac{1}{|\mathcal{D}_l|}\sum_{z\in\mathcal{D}_l}\mathcal{F}_{\Theta_i}(z)\right)}_{\text{ProtoDist}(x,\mathcal{D}_{c_{i-1}+1:c_i})},
$$

$$(1)$$

---

**Algorithm 1** Training the CIL model in Phase $i$ ($i \geq 1$)

1: **Input:** New data $\mathcal{D}_{c_{i-1}+1:c_i}$, old exemplars $\mathcal{E}_{1:c_{i-1}}$, old model $\Theta_{i-1}$, and pre-trained policy fucntion $\pi_\phi$
2: **Output:** New model $\Theta_i$, new exemplars $\mathcal{E}_{1:c_i}$
3: Initialize $\Theta_i$ with $\Theta_{i-1}$;
4: Observe $s_i$ and produce $(\beta_i, \gamma_i) \sim \pi_\phi(s_i)$;
5: **for** epochs **do**
6: $\quad$ Create $\{S_m(x)\}_{m=1}^{c_{i-1}}$ using Eq. 1 and set $\mathcal{P} = \varnothing$;
7: $\quad$ **while** $\mathcal{P} == \varnothing$ **do**
8: $\quad\quad$ Sample $\mathcal{U}$ from the free image stream;
9: $\quad\quad$ Select placebos $\mathcal{P} \subset \mathcal{U}$ using Eq. 2;
10: $\quad\quad$ **for** iterations **do**
11: $\quad\quad\quad$ Sample mini-batches $\boldsymbol{p}$, $\boldsymbol{d}$, and $\boldsymbol{e}$;
12: $\quad\quad\quad$ Calculate loss $\mathcal{L}$ using Eq. 3;
13: $\quad\quad\quad$ Update the CIL model $\Theta_i \leftarrow \Theta_i - \alpha \nabla_{\Theta_i} \mathcal{L}$;
14: $\quad\quad\quad$ Update knockoffs $\mathcal{P} := \mathcal{P} \setminus \boldsymbol{p}$;
15: Select new exemplars $\mathcal{E}_{1:c_i}$ using herding.

---

**Algorithm 2** Pre-training the policy functions

1: **Input:** Data $\mathcal{D}$ for generating pseudo CIL tasks.
2: **Output:** Policy function $\pi_\phi$.
3: Initialize $\phi$;
4: **for** $w$ RL epochs **do**
5: $\quad$ **for** $g$ **in** $1, ..., G$ **do**
6: $\quad\quad$ Create a new pseudo task $\mathcal{T}_g$ using $\mathcal{D}$;
7: $\quad\quad$ **for** $h$ **in** $1, ..., H$ **do**
8: $\quad\quad\quad$ Initialize classification model $\Theta_0$;
9: $\quad\quad\quad$ **for** $i$ **in** $0, ..., N$ **do**
10: $\quad\quad\quad\quad$ **if** $i == 1$ **do**
11: $\quad\quad\quad\quad\quad$ Train $\Theta_0$ as a conventional classification task;
12: $\quad\quad\quad\quad$ **else do**
13: $\quad\quad\quad\quad\quad$ Observe $s_i$ and train $\Theta_i$ using **Algorithm 1**;
14: $\quad\quad\quad\quad$ Compute validation accuracy $r_i$;
15: $\quad\quad\quad$ Compute $R_g^h = \sum_{i=0}^{N} r_i$ and update $b$;
16: $\quad$ Compute $\nabla_\phi J(\phi)$ (Eq. 6) and update $\phi$ (Eq. 7).

---

where $x$ denotes an unlabeled input image, $\mathcal{F}_{\Theta_i}(\cdot)$ denotes the encoder of $\Theta_i$ and $\cos(\cdot, \cdot)$ denotes the cosine similarity. $\mathrm{ProtoDist}(x, \mathcal{E}_m)$, $\mathrm{ProtoDist}(x, \mathcal{E}_{-m})$, and $\mathrm{ProtoDist}(x, \mathcal{D}_{c_{i-1}+1:c_i})$ are the (average) cosine similarities in feature space between $x$ and of the $m$-th old class prototype, other old class prototypes, new class prototypes, respectively. $\beta_i$ and $\gamma_i$ are two hyperparameters to balance the above three parts. If we set $\beta_i = \gamma_i = 1$, $\mathcal{S}_m(x)$ will be a simple and heuristic evaluation function, which is also effective according to our empirical results (see Section 5). To make the evaluation function more efficient, we propose to train a policy using RL to generate phase-specific $\beta_i$ and $\gamma_i$, and the details are given in Section 4.2.

**Selecting the placebos.** At the beginning of the $i$-th phase, we empty the placebo buffer $\mathcal{P}$. Whenever $\mathcal{P}$ is empty, we load a batch of unlabeled samples $\mathcal{U}$ from the free image stream, and add $K$ placebos for each old class to $\mathcal{P}$ as follows,

$$\mathcal{P} := \{x_k\}_{k=1}^{c_{i-1}K} = \arg\max_{x_k \in \mathcal{U}} \sum_{m=1}^{c_{i-1}} \sum_{k=1}^{K} \mathcal{S}_m(x_k). \tag{2}$$

**Calculating the loss with the placebos.** After selecting the placebos, we sample a batch of new class data $\boldsymbol{d} \subset \mathcal{D}_{c_{i-1}+1:c_i}$, a batch of old class exemplars $\boldsymbol{e} \subset \mathcal{E}_{1:c_0}$, and a batch of placebos $\boldsymbol{p} \subset \mathcal{P}$. We calculate the overall loss as follows,

$$\mathcal{L} = \mathcal{L}_{\mathrm{CE}}(\Theta_i; \boldsymbol{d} \cup \boldsymbol{e}) + \lambda \mathcal{L}_{\mathrm{KD}}(\Theta_{i-1}, \Theta_i; \boldsymbol{p} \cup \boldsymbol{e}), \tag{3}$$

where $\mathcal{L}_{\mathrm{CE}}$ and $\mathcal{L}_{\mathrm{KD}}$ denote the CE loss and KD loss, respectively. $\lambda$ is a hyperparameter to balance the two losses (Rebuffi et al., 2017). For feature distillation methods (Hou et al., 2019; Douillard et al., 2020), the KD issue is not a major limitation. However, we empirically find our placebos are still effective for retaining the old knowledge; thus, we use $\boldsymbol{p} \cup \boldsymbol{d} \cup \boldsymbol{e}$ to compute the feature KD loss.

**Memory management.** We need a small amount of additional memory to store the unlabeled image batch $\mathcal{U}$. To obtain this memory, we randomly delete $|\mathcal{U}|$ samples from the new class data $\mathcal{D}_{c_{i-1}:c_i}$. The empirical results show the above step hardly degrades performance (see Section 5.2 **Ablation study**). During training, we delete $\boldsymbol{p}$ from $\mathcal{P}$ immediately after calculating the loss. When we run out of the selected placebos in $\mathcal{P}$, we load another batch of unlabeled data $\mathcal{U}$ from the free image stream and repeat the selecting steps to get new placebos $\mathcal{P}$.

## 4.2 LEARNING THE POLICY FOR EVALUATION FUNCTIONS

Because the ratio between old and new classes changes significantly when the number of phases increases, the evaluation functions should also change accordingly. To achieve this, we design an

RL system and train a policy to build phase-specific evaluation functions. The overall objective is to maximize the cumulative evaluation accuracy across all incremental phases. However, this is not compatible with the standard CIL protocol (Rebuffi et al., 2017) where neither past nor future data are accessible. To tackle this issue, we sample pseudo CIL tasks and optimize the policy on them. In the following, we first introduce the formulation of our RL system, including the definitions of the states, actions, and rewards. Then we show how to create the pseudo CIL tasks and train the policy. We summarize the learning process of the policy in Algorithm 2.

**Formulations of the RL system.** *States.* We define the states based on the following two requirements. (1) The states should be transferable between different CIL tasks, e.g., from a small-scale CIL task including 100 classes to a large one including $1,000$ classes. It is because we need to transfer the policy learned from pseudo CIL tasks to the target task. So the states should also be transferable. (2) The state should be distinct in each incremental phase. This is to enable the state variable to represent a specific forgetting degree at each phase. To fulfill these requirements, we define the state in the $i$-th incremental phase as $s_i = \frac{c_i - c_{i-1}}{c_{i-1}}$, where $(c_i - c_{i-1})$ is the number of new classes we observe in the $i$-th phase, and $c_{i-1}$ is the number of old classes we have seen in all previous phases. *Actions.* We use a vector that consists of the phase-specific hyperparameters in the evaluation function as the action, i.e., $\boldsymbol{a}_i = (\beta_i, \gamma_i)$. When we take action $\boldsymbol{a}_i$, we build evaluation functions $\{S_m(u)\}_{m=1}^{c_{i-1}}$ with $(\beta_i, \gamma_i)$, and deploy them in our CIL pipeline. *Rewards.* The objective of CIL is to train a model that is efficient to recognize all classes seen so far. So it is intuitive to use the validation accuracy as the reward in each phase. In the $i$-th phase, the objective of the RL system is to maximize the cumulative reward, i.e., $R = \sum_{i=0}^{N} r_i$, where $r_i$ denotes the validation accuracy in the $i$-th phase.

**Creating pseudo CIL tasks.** As we need to compute the cumulative rewards, we need to access all training and validation data of the pseudo CIL tasks. Thus, an intuitive solution is using $\mathcal{D}_{1:c_0}$ (available in the 0-th phase) to generate pseudo CIL tasks. Based on the CIL protocol (Douillard et al., 2020; Hou et al., 2019; Hu et al., 2021; Liu et al., 2021a), $\mathcal{D}_{1:c_0}$ contains half of the classes of the whole dataset, e.g., $500$ classes on ImageNet-1k, which supplies enough data to build downsized CIL tasks. When building the tasks, we randomly choose $10\%$ training samples of each class (from $\mathcal{D}_{1:c_0}$) to compose a pseudo validation set (note that we are not allowed to use the original validation set in training). To train a more robust policy we generate pseudo CIL tasks with different class orders and different numbers of phases.

**Training with REINFORCE.** We elaborate the steps of learning policy $\pi_\phi$ (parameterized by $\phi$) in the following. The goal is to optimize $\phi$ by maximizing the expected cumulative reward $J(\phi)$. We denote any pseudo CIL task as $\mathcal{T}$ and its cumulative reward as $R$, and have,

$$J(\phi) = \mathbb{E}_{\mathcal{T}} \mathbb{E}_{\pi_\phi}[R]. \tag{4}$$

*Policy gradient estimation.* According the policy gradient theorem (Williams, 1992) we can compute the gradients for $J(\eta, \phi)$ as follows,

$$\nabla_\phi J(\phi) = \mathbb{E}_{\mathcal{T}} \left[ \sum_{i=1}^{N} \mathbb{E}_{\pi_\phi} [\nabla_\phi \log \pi_\phi(\boldsymbol{a}_i|s_i) R] \right]. \tag{5}$$

Following the REINFORCE algorithm (Williams, 1992), we replace the expectations $\mathbb{E}_{\mathcal{T}}[\cdot]$ and $\mathbb{E}_{\pi_\phi}[\cdot]$ with sample averages (i.e., the Monte Carlo method (Hammersley, 2013)). Specifically, in each RL epoch, we create $G$ pseudo tasks and run each task for $H$ times. Thus we can derive the empirical approximation of $\nabla_\phi J(\phi)$ as,

$$\frac{1}{GH} \sum_{g=1}^{G} \sum_{h=1}^{H} \sum_{i=1}^{N} \nabla_\phi \log \pi_\phi(\boldsymbol{a}_i|s_i)(R_g^h - b), \tag{6}$$

where $R_g^h$ denotes the $g$-th reward for the $h$-th pseudo task $\mathcal{T}_h$, and $b$ denotes the baseline function— the moving average of previous rewards. Using this baseline function is a common trick in RL to reduce the variance of estimated policy gradients (Rennie et al., 2017; Zoph & Le, 2017).

*Updating policy parameters.* We update $\phi$ (i.e., the parameters in $\pi_\phi$) in each RL epoch using gradient ascent (Xu & Zhu, 2018; Zoph & Le, 2017):

$$\phi := \phi + \eta \nabla_\phi J(\phi), \tag{7}$$

where $\eta$ is the learning rate. We iterate this update for $w$ RL epochs in total.

| Method | 20 exemplars/class | | 10 exemplars/class | | 5 exemplars/class | |
|---|---|---|---|---|---|---|
| | Average | Last | Average | Last | Average | Last |
| LwF | 53.19 | 43.18 | 45.96 | 34.10 | 35.41 | 24.91 |
| *w/* ours | 59.29 +6.10 | 49.64 +6.46 | 53.48 +7.52 | 38.03 +3.93 | 41.67 +6.26 | 28.60 +3.69 |
| iCaRL | 57.12 | 47.49 | 53.43 | 41.49 | 43.73 | 34.33 |
| *w/* ours | 61.17 +4.05 | 50.96 +3.47 | 59.32 +5.89 | 46.48 +4.99 | 51.19 +7.46 | 39.29 +4.96 |
| LUCIR | 63.17 | 53.71 | 60.50 | 49.08 | 51.36 | 39.37 |
| *w/* ours | 65.48 +2.31 | 56.77 +3.06 | 64.93 +3.89 | 55.54 +6.46 | 63.01 +11.65 | 53.09 +13.72 |
| LUCIR+AANets | 66.72 | 57.77 | 65.46 | 55.17 | 60.28 | 48.23 |
| *w/* ours | 67.33 +0.61 | 59.32 +1.55 | 65.51 +0.05 | 55.42 +0.25 | 64.10 +3.82 | 53.41 +5.18 |
| POD+AANets | 66.12 | 55.27 | 61.12 | 48.83 | 53.81 | 42.93 |
| *w/* ours | 67.47 +1.35 | 58.91 +3.64 | 64.56 +3.44 | 52.60 +3.77 | 60.35 +6.54 | 48.53 +5.60 |

Table 1: Evaluation results (%) on 5-Phase CIFAR-100 using POD+AANets (Liu et al., 2021a), LUCIR+AANets (Liu et al., 2021a), LUCIR (Hou et al., 2019), iCaRL (Rebuffi et al., 2017), and LwF (Li & Hoiem, 2016) *w/* and *w/o* our PlaceboCIL plugged in. "Average" and "Last" denote the average accuracy over five phases and the last-phase (5-th) accuracy, respectively.

## 5 EXPERIMENTS

We evaluate our method on three CIL benchmarks (CIFAR-100, ImageNet-Subset, and ImageNet-1k) and achieve clear and consistent improvements. Below we describe the datasets and implementation details, followed by results and analyses including the comparison to the state-of-the-art methods, an ablation study and the visualization of our placebo samples.

### 5.1 DATASETS AND IMPLEMENTATION DETAILS

**Datasets and free image streams.** We use three benchmarks based on two datasets, CIFAR-100 (Krizhevsky et al., 2009) and ImageNet-1k (Russakovsky et al., 2015), and follow the same data splits in related works (Douillard et al., 2020; Rebuffi et al., 2017; Liu et al., 2021a). There are two CIL settings on ImageNet-1k: ImageNet-Subset using a subset of 100 classes, and ImageNet-1k using the full set of $1,000$ classes. The 100-class data for the ImageNet-Subset are sampled from ImageNet-1k. For CIFAR-100, we use ImageNet-1k as the free image stream. For ImageNet-Subset, we use a 900-class subset of ImageNet-1k, which is the complement of ImageNet-Subset in ImageNet-1k. For ImageNet-1k, we use a $1,000$-class subset of ImageNet-21k (Deng et al., 2009) without any overlapping class (totally different super classes from those in ImageNet-1k).

**Configurations.** Following Hou et al. (2019); Douillard et al. (2020), we use a 32-layer ResNet (Rebuffi et al., 2017) for CIFAR-100 dataset and an 18-layer ResNet (He et al., 2016) for ImageNet datasets. We follow the benchmark protocol used in Douillard et al. (2020); Hou et al. (2019). The 0-th phase model is trained on the data of half of the classes. Then, it learns the remaining classes evenly in the subsequent $N$ phases. The number of exemplars for each class is 20 in the default setting. The training batch size is 128. $|\mathcal{U}|$ and $K$ are set as $1,000$ and $200$, respectively.

### 5.2 RESULTS AND ANALYSES

**Results on five baselines.** Table 1 shows the average and last-phase accuracy for five baselines (i.e., POD+AANets, LUCIR+AANets (Liu et al., 2021a), LUCIR (Hou et al., 2019), iCaRL (Rebuffi et al., 2017), and LwF (Li & Hoiem, 2016). From the table, we make the following observations. 1) Using our PlaceboCIL boosts the performance of the baselines clearly and consistently in all settings, indicating that our method is general and robust. 2) When the number of exemplars decreases, the improvement brought by our method becomes more significant. For example, the average last-phase accuracy improvement increases from 3.63 to 6.63 percentage points when the number of exemplars per class decreases from 20 to 5. It reveals that the superiority of our method is more obvious when the forgetting problem is more serious due to a tighter memory budget (more challenging for CIL). 3) Our PlaceboCIL can boost the performance of all KD regularization, i.e., not only for prediction logits-based KD (Rebuffi et al., 2017) but also for feature-based KD (Hou et al., 2019).

| Method | CIFAR-100 | | | ImageNet-Subset | | | ImageNet-1k | | |
|---|---|---|---|---|---|---|---|---|---|
| | $N$=5 | 10 | 25 | 5 | 10 | 25 | 5 | 10 | 25 |
| TPCIL (Tao et al., 2020) | 65.34 | 63.58 | – | 76.27 | 74.81 | – | 64.89 | 62.88 | – |
| PODNet (Douillard et al., 2020) | 64.83 | 63.19 | 60.72 | 75.54 | 74.33 | 68.31 | 66.95 | 64.13 | 59.17 |
| DDE (Hu et al., 2021) | 65.42 | 64.12 | – | 76.71 | 75.41 | – | 66.42 | 64.71 | – |
| GeoDL (Simon et al., 2021) | 65.14 | 65.03 | 63.12 | 76.63 | 75.40 | 71.43 | 65.23 | 64.79 | 60.97 |
| DER (Yan et al., 2021)[†] | 68.65 | 67.48 | 66.18 | – | 78.20 | – | – | – | – |
| POD-AANets (Liu et al., 2021a) | 66.12 | 64.11 | 62.12 | 76.63 | 75.40 | 71.43 | 67.60 | 64.79 | 60.97 |
| *w/* PlaceboCIL (ours) | **67.47** | **65.70** | **64.53** | **78.14** | **77.08** | **75.50** | **68.61** | **65.69** | **61.76** |

[†]DER (Yan et al., 2021) requires additional memory to store the store all learned encoders (in all phases).

Table 2: Average accuracy (%) across all phases using the state-of-the-art method (POD+AANets) *w/* and *w/o* our PlaceboCIL plugged in. The upper block is for recent CIL methods.

| No. | Setting | LwF | | iCaRL | | LUCIR | | LUCIR+AANets | |
|---|---|---|---|---|---|---|---|---|---|
| | | Average | Last | Average | Last | Average | Last | Average | Last |
| 1 | Baseline | 53.19 | 43.18 | 57.12 | 47.49 | 63.17 | 53.71 | 66.72 | 57.77 |
| 2 | PlaceboCIL (budget) | 59.10 | 48.28 | 61.09 | 50.81 | 65.14 | 56.95 | 67.31 | 59.26 |
| 3 | PlaceboCIL (non-budget) | 59.29 | 49.64 | 61.17 | 50.96 | 65.48 | 56.77 | 67.33 | 59.32 |
| 4 | Overlapping | 58.95 | 48.71 | 62.15 | 52.62 | 65.73 | 57.26 | 67.48 | 59.06 |
| 5 | Non-overlapping | 58.59 | 49.28 | 61.52 | 51.70 | 65.48 | 56.99 | 67.01 | 58.53 |
| 6 | New data | 54.06 | 43.94 | 57.70 | 47.51 | 64.21 | 53.98 | 66.69 | 57.33 |
| 7 | Old data (upper bound) | 61.41 | 51.16 | 66.64 | 58.03 | 67.02 | 58.98 | 68.82 | 61.52 |
| 8 | *w/o* RL | 57.06 | 46.12 | 60.27 | 50.57 | 63.83 | 55.33 | 66.91 | 58.88 |
| 9 | *w/* Transferred RL | 58.36 | 48.98 | 61.58 | 52.51 | 65.04 | 56.48 | 67.18 | 57.84 |
| 10 | CE loss on placebos | 56.23 | 45.74 | 56.40 | 44.70 | 65.03 | 55.84 | 66.05 | 56.02 |
| 11 | Higher confidence | 56.98 | 46.20 | 60.43 | 49.36 | 63.60 | 55.50 | 66.97 | 58.12 |
| 12 | Random placebos | 50.99 | 40.22 | 56.27 | 46.64 | 64.16 | 55.40 | 66.23 | 57.22 |

Table 3: Ablation results (%) on CIFAR-100, $N$=5. "Average" and "Last" denote the average accuracy over all phases and the last-phase accuracy, respectively. The four blocks show the results of baselines, using different free image streams, using RL (or not), and using placebos to compute CE loss, respectively. Please refer to more details in Section 5.2 **Ablation study**.

**Comparison to the state-of-the-art.** Table 2 shows the results of state-of-the-art methods and our best model (taking PlaceboCIL as a plug-in module in the top method (Liu et al., 2021a)). We can see that using our PlaceboCIL outperforms previous methods except for DER (Yan et al., 2021), which requires lots of additional memory to store the encoders of all previous phases. Intriguingly, we find that we can surpass others more when the number of phases is larger—where there are more serious forgetting problems. For example, when $N$=25, we improve POD-AANets by $4.1\%$ on the ImageNet-Subset, while this number is only $1.5\%$ when $N$=5 (which is an easier setting with more saturated results). This reflects the encouraging efficiency of our method for reducing the forgetting of old class knowledge in CIL models.

**Ablation study.** Table 3 shows an ablation study. ***First block: baselines.*** Row 1 shows baselines. Rows 2 and 3 show the baselines with our PlaceboCIL as a plug-in module. Row 2 "budget" means the combined size of episodic memory and the placebo buffer matching to the episodic memory size in other comparable methods. To this end, we randomly delete $|\mathcal{U}|$ samples from the new class data $\mathcal{D}_{c_{i-1}+1:c_i}$ to obtain the buffer memory storing the unlabeled data batch $\mathcal{U}$. Row 3 "non-budget" means we are allowed to use a little additional memory to save the unlabelled data batch $\mathcal{U}$ without deleting other samples. Comparing Row 3 to Row 2, we can see that the performance of our PlaceboCIL only reduces 0.16 percentage points on average under the "budget" setting. The reason is that the amount of new data is much larger than that of old samples, and removing a small batch of new data does not hurt performance much (Liu et al., 2021b). If not specified, our reported results in other tables are from the setting of "non-budget".
***Second block: different free data streams.*** Rows 4-7 show the ablation results for the following free data streams. (1) "Overlapping" means including samples from the overlapping classes between CIFAR-100 and ImageNet. (2) "Non-overlapping" means using only the samples of non-overlapping classes between CIFAR-100 and ImageNet (more realistic than "Overlapping"). (3) "New data" means using only the current-phase new class data (i.e., without using any free data stream) as

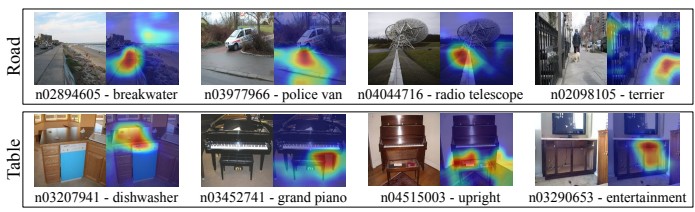 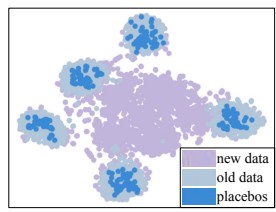

(a) Selected placebos and GradCAM visualization  (b) t-SNE visualization

Figure 3: (a) The selected placebos for two classes ("road" and "table") and their activation maps using GradCAM (Selvaraju et al., 2017) on CIFAR-100 ($N$=5). The free image stream is non-matching ImageNet-1k. (b) The t-SNE (Maaten & Hinton, 2008) results on CIFAR-100 ($N$=5). For clear visualization. We randomly pick five new classes and five old classes. The purple, light blue, and dark blue points denote the new data, old data, and the selected placebos, respectively.

candidates to select placebos. (4) "Old data" means the original old class data are all accessible when computing KD loss (i.e., upper bound of KD effect). Please note that in (1) and (2), two classes are considered "overlapping" if their classes or super-classes overlap. For example, "n02640242 - sturgeon" in ImageNet-1k is regarded as an overlapping class of the "fish" in CIFAR-100, because they overlap at the level of super-class (i.e., "fish"). When comparing Row 5 with Row 3, we can find that our method is robust to the change of data streams: even if all overlapping classes are removed, our method can still achieve the same-level performance. Comparing Row 6 with Row 3, we can get the clear sense that using additional unlabeled data is definitely helpful. Comparing Row 7 with Row 3, we see that our method achieves comparable results to the upper bound.

***Third block: RL.*** Row 8 is for using heuristic evaluation functions ($\beta_i=\gamma_i=1$) instead of RL. Row 9 uses the policy transferred from ImageNet-Sub, which means on the target CIL dataset, there is no RL training. Comparing Row 8 with Row 3 shows that using the RL algorithm successfully boosts the model performance. Comparing Row 9 with Row 3, we are happy to see that the RL policy learned is generalizable and can be transferred to the placebo selection on other datasets.

***Fourth block: CE loss.*** Row 10 assigns pseudo labels to the selected placebos and compute CE loss instead of KD loss in CIL. Comparing Row 10 with Row 3 shows using placebo data to compute CE loss significantly hurts the performance. The reason is that soft labels (used for KD) contain not only useful logits but noisy logits. When computing CE loss, soft labels are converted to hard (one-hot) labels that could be completely wrong labels—making the model overfitted to noisy labels.

***Fifth block: Different Placebo Selection Strategies.*** Row 11 chooses the placebos with higher confidence (Lee et al., 2019). Row 12 chooses the placebos randomly like (Zhang et al., 2020). Comparing Row 11 (Row 12) with Row 3 shows using our RL-learned placebo selection strategy achieves better performance. The reason is that our evaluation functions change accordingly when the number of phases increases and the ratio between old and new classes changes.

**Visualization results.** Figure 3 (a) demonstrates the activation maps visualized by Grad-CAM for the placebos of two old classes on CIFAR-100 ("road" and "table"). ImageNet-1k is the free data stream. We can see that the selected placebos contain the parts of "road" and "table" even though their original labels (on ImageNet-1k) are totally different classes. While this is not always the case, our method seems to find sufficiently related images to old classes that activate the related neurons for old classes ("road" and "table"). To illustrate that, figure 3 (b) shows t-SNE results for placebos, old data (not visible during training), and new data. We can see that the placebo samples locate near the old data and far away from the new data. This is why placebos can recall the old knowledge and do not harm the learning of new classes.

# 6 CONCLUSIONS

We proposed a novel method PlaceboCIL for tackling the forgetting problem in CIL tasks. PlaceboCIL selects high-quality placebo data from free data streams, and use them to improve the effect of KD between the models learned in adjacent phases, while not harming the learning of new classes in the new model. We designed an RL algorithm to make the selection of placebos more adaptive in different phases. Extensive experiments on multiple baselines show that our method is general and efficient. We believe it is promising to leverage unlabeled or free data in solving the data scarce problems in continual learning tasks, and will continue to explore this direction.

ETHICS STATEMENT

**Computational costs.** RL-based methods require intensive usage of computing resources, which is not climate-friendly. It calls for future research into proposing more effective RL training strategies that can speed up the training.

**Privacy issues.** Keeping old class exemplars has the issue of data privacy. This calls for future research that explicitly forgets or mitigates the identifiable feature of the data.

**Licenses.** We use the open-source code for the following papers: AANets (Liu et al., 2021a), iCaRL (Rebuffi et al., 2017), Mnemonics (Liu et al., 2020a), and PODNet (Douillard et al., 2020). They are licensed under the MIT License.

**Datasets.** We use two datasets in our paper: CIFAR-100 (Krizhevsky et al., 2009) and ImageNet-1k (Russakovsky et al., 2015). The data for both datasets are downloaded from their official websites and allowed to use for non-commercial research and educational purposes.

REPRODUCIBILITY STATEMENT

We include the details of reproducing our experiments in Section 5.1 (main paper), Section C (appendix) and Section D (appendix). We also promise to release the code if this paper is accepted.

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

These appendices include the additional experiment results (§A), benchmark protocols (§B), network architecture details (§C), training configurations (§D), equal-size split results (§E), learning curve area results (§F), ablation results for different unlabeled data sources (§G), hardware information (§H), and additional plots (§I).

## A    ADDITIONAL EXPERIMENT RESULTS

This is supplementary to Section 5.2 "**Results on five baselines**." In Table A1, we supplement the average and last-phase accuracy on five baselines (i.e., POD+AANets, LUCIR+AANets (Liu et al., 2021a), LUCIR (Hou et al., 2019), iCaRL (Rebuffi et al., 2017), and LwF (Li & Hoiem, 2016) on CIFAR-100 ($N$=25). We can observe that using our PlaceboCIL boosts the performance of the baselines clearly and consistently in all settings.

| Method | 20 exemplars/class | | 10 exemplars/class | | 5 exemplars/class | |
|---|---|---|---|---|---|---|
| | Average | Last | Average | Last | Average | Last |
| iCaRL | 48.22 | 39.56 | 40.40 | 33.14 | 32.53 | 19.35 |
| *w/* ours | 52.46 +4.24 | 42.01 +2.45 | 42.48 +2.08 | 33.20 +0.06 | 36.40 +3.87 | 22.70 +3.35 |
| LUCIR | 57.54 | 48.32 | 49.78 | 40.63 | 45.64 | 32.16 |
| *w/* ours | 63.66 +6.12 | 53.77 +5.45 | 61.22 +11.44 | 50.84 +10.21 | 59.82 +14.18 | 48.83 +16.67 |
| LUCIR+AANets | 64.29 | 52.99 | 59.75 | 48.19 | 55.26 | 39.60 |
| *w/* ours | 65.96 +1.67 | 57.67 +4.68 | 63.60 +3.85 | 52.67 +4.48 | 63.01 +7.75 | 51.88 +12.28 |
| POD+AANets | 62.12 | 51.73 | 53.31 | 42.83 | 45.09 | 32.97 |
| *w/* ours | 64.53 +2.41 | 54.95 +3.22 | 58.75 +5.44 | 46.83 +4.00 | 47.86 +2.77 | 33.93 +0.96 |

Table A1: Supplementary to Table 1. Evaluation results (%) on 25-Phase CIFAR-100 using POD+AANets (Liu et al., 2021a), LUCIR+AANets (Liu et al., 2021a), LUCIR (Hou et al., 2019), and iCaRL (Rebuffi et al., 2017) *w/* and *w/o* our PlaceboCIL plugged in. "Average" and "Last" denote the average accuracy over five phases and the last-phase (25-th) accuracy, respectively.

## B    BENCHMARKS PROTOCOL.

We follow the benchmark protocol used in Douillard et al. (2020); Hou et al. (2019); Liu et al. (2021a; 2020a). Given a dataset, the initial model (in the 0-th phase) is trained on the data of half of the classes. Then, it learns the remaining classes evenly in the subsequent $N$ phases. The learned model in each phase is evaluated on the test set containing all seen classes. In the tables, we report average accuracy over all phases and the last-phase accuracy.

## C    NETWORK ARCHITECTURE DETAILS

This is supplementary to Section 5.1 "**Network architectures and configurations..**" Following (Hou et al., 2019; Liu et al., 2021a; Rebuffi et al., 2017; Wu et al., 2019), we use a 32-layer ResNet (Rebuffi et al., 2017) for CIFAR-100 and an 18-layer ResNet (He et al., 2016) for ImageNet-1k. Please note that it is standard to use a shallower ResNet for ImageNet-1k. The 32-layer ResNet consists of 1 initial convolution layer and 3 residual blocks (in a single branch). Each block has 10 convolution layers with $3 \times 3$ kernels. The number of filters starts from 16 and is doubled every next block. After these 3 blocks, there is an average-pooling layer to compress the output feature maps to a feature embedding. The 18-layer ResNet follows the standard settings in (He et al., 2016). We deploy AANets using the same parameters as its original paper (Liu et al., 2021a).

## D    TRAINING CONFIGURATIONS

This is supplementary to Section 5.1 "**Network architectures and configurations..**" The training of the classification model $\Theta$ exactly follows the uniform setting in (Douillard et al., 2020; Hou et al.,

2019; Liu et al., 2021a; 2020a). On CIFAR-100 (ImageNet-Subset/1k), we train it for 160 (90) epochs in each phase, and divide the learning rate by 10 after 80 (30) and then after 120 (60) epochs. If the baseline is POD+AANets (Liu et al., 2021a), we fine-tune the model for 20 epochs using only exemplars. We apply different forms of distillation losses on different baselines: (1) if the baselines are LwF (Li & Hoiem, 2016) and iCaRL (Rebuffi et al., 2017), we use the softmax KL divergence loss; (2) if the baselines are LUCIR (Hou et al., 2019) and LUCIR+AANets (Liu et al., 2021a), we use the cosine embedding loss (Hou et al., 2019); and (3) if the baseline is PODNet (Douillard et al., 2020) and POD+AANets (Liu et al., 2021a), we use pooled outputs distillation loss (Douillard et al., 2020).

## E  EQUAL-SIZE SPLIT RESULTS

This is supplementary to Section 5.2 "**Ablation study**." In Table A2, we supplement the results using the equal-size split on CIFAR 100, 10-phase (10 classes/phase, 20 exemplars/class). We pre-train the RL policy on ImageNet-Subset. We can observe that using placebos selected by heuristic evaluation functions ($\beta_i = \gamma_i = 1$) consistently improves the results on three baselines. Using the RL algorithm can further boost performance. The observations are similar to using the original setting in the main paper.

| No. | Setting | LwF | | iCaRL | | LUCIR | |
|-----|---------|-----|-----|-------|-----|-------|-----|
| | | Average | Last | Average | Last | Average | Last |
| 1 | Baseline | 53.85 | 41.00 | 59.70 | 45.29 | 56.71 | 42.78 |
| 2 | PlaceboCIL (ours) | 57.68 | 42.44 | 62.32 | 46.52 | 57.89 | 44.08 |
| 3 | PlaceboCIL (ours, w/o RL) | 56.31 | 41.76 | 61.05 | 46.02 | 57.01 | 43.13 |

Table A2: Supplementary to Table 3. Ablation results (%) using equal-size split on CIFAR-100, 10-phase (10 classes/phase).

## F  LEARNING CURVE AREA RESULTS

This is supplementary to Section 5.2 "**Ablation study**."In Table A3, we show the Learning Curve Area ($LCA_{10}$) (Chaudhry et al., 2019) on CIFAR-100, 5-phase. We can observe that our PlaceboCIL achieves much better $LCA_{10}$ compared to the baselines. It shows that our PlaceboCIL improves the ability to learn new knowledge.

| No. | Setting | LwF | iCaRL | LUCIR | LUCIR +AANets |
|-----|---------|-----|-------|-------|---------------|
| 1 | Baseline | 40.02 | 47.79 | 52.06 | 56.44 |
| 2 | PlaceboCIL (ours) | 50.82 | 55.24 | 57.48 | 59.69 |

Table A3: Supplementary to Table 3. Learning Curve Area ($LCA_{10}$) (Chaudhry et al., 2019) on CIFAR-100, 5-phase.

## G  ABLATION RESULTS FOR DIFFERENT UNLABELED DATA SOURCES

This is supplementary to Section 5.2 "**Ablation study**." We believe that the key to success is the design of our method instead of the choice of unlabeled data sources. To verify this, we provide the results for a new ablative setting: "using random unlabeled data" (to compute the distillation loss) in Table A4. We can observe that no matter what unlabeled data sources we use, our method consistently performs better than using random unlabeled data.

## H  HARDWARE INFORMATION

We run our experiments using GPU workstations as follows,

| No. | Setting | LwF | | iCaRL | | LUCIR | | LUCIR+AANets | |
|---|---|---|---|---|---|---|---|---|---|
| | | Average | Last | Average | Last | Average | Last | Average | Last |
| 1 | Baseline | 53.19 | 43.18 | 57.12 | 47.49 | 63.17 | 53.71 | 66.72 | 57.77 |
| 2 | PlaceboCIL (ours, all) | 59.29 | 49.64 | 61.17 | 50.96 | 65.48 | 56.77 | 67.33 | 59.32 |
| 3 | PlaceboCIL (ours, overlapping) | 58.95 | 48.71 | 62.15 | 52.62 | 65.73 | 57.26 | 67.48 | 59.06 |
| 4 | Random unlabeled data (all) | 50.99 | 40.22 | 56.27 | 46.64 | 64.16 | 55.45 | 66.23 | 57.22 |
| 4 | Random unlabeled data (overlapping) | 50.80 | 40.94 | 55.70 | 46.47 | 64.23 | 54.68 | 66.58 | 57.08 |

Table A4: Supplementary to Table 3. Ablation results (%) on CIFAR-100, $N$=5. "Average" and "Last" denote the average accuracy over all phases and the last-phase accuracy, respectively. "All" denotes using all data from ImageNet, and "overlapping" means including samples from the overlapping classes between CIFAR-100 and ImageNet.

- **CPU**: 1x AMD EPYC 7502P 32-Core Processor
- **GPU**: 4x NVIDIA Quadro RTX 8000, 48 GB GDDR6
- **Memory**: 1024 GiB = 8x 128GiB, DDR4, 3200 MHz, ECC

# I ADDITIONAL PLOTS

This is supplementary to Section 5.2 "**Compared to the state-of-the-art**." In Figures A1, we present the phase-wise accuracy obtained on CIFAR-100 and ImageNet-Subset. "Upper Bound" shows the results of joint training with all previous data accessible in every phase. We can observe that our method achieves the highest accuracy in almost every phase of different settings.

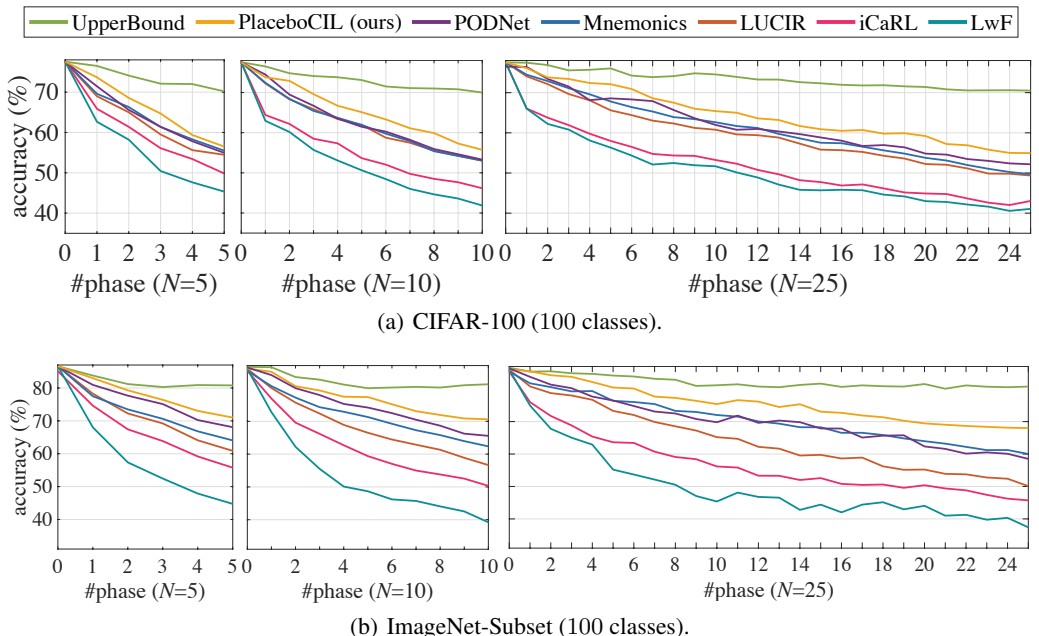

(a) CIFAR-100 (100 classes).

(b) ImageNet-Subset (100 classes).

Figure A1: Supplementary to Table 2. Phase-wise accuracies (%) on CIFAR-100 and ImageNet-Subset.

