# OpenReview forum: "Wakening Past Concepts without Past Data: Class-incremental Learning from Placebos"
_ICLR.cc/2022/Conference — ICLR 2022 Submitted_

### Official Review · Reviewer_Dwjy · 2021-10-20

**Correctness:** 1
**Technical Novelty And Significance:** 2
**Empirical Novelty And Significance:** 2
**Recommendation:** 5
**Confidence:** 4

**Main Review:**

**Strengths**
- S1: Novel idea of using unlabeled data for knowledge distillation for class incremental learning (CIL).
- S2: Compelling empirical performance with large datasets.
- S3: Clear presentations (i.e., figures, texts and equations)

**Weaknesses**
- W1: Out of protocol of CIL -- using large unlabeled data requires large supplementary data source and require additional computational cost, which may limits the applicability of the CIL method. The rationale behind this claim is that the CIL is for the realistic recognition scenario that the data is given in a stream and the model update should be on the fly, adaptively to the new data. If the computation of incremental update step is large, the benefit of the CIL could disappear.
- W2: Large unlabeled source may or may not contain sufficient information (i.e., even the future information) depending on the choice of source of unlabeled data -- Google searched image could be the superset of the ImageNet-1k as ImageNet dataset is collected from Google search. Thus, the choice of unlabeled data might be the key to success of the proposed method, not the design of the method. Lack of study about the source is a weak point of significance of the proposed method
- W3: The evaluation protocol of using small memory budget is largely beneficial to the proposed method. As the proposed method is using large unlabeled data as an extra source of information, the small memory budget is easily compensated by the proposed method while other methods inevitably performs poorly due to lack of information source.
- W4: Similar to W3, the combined size of episodic memory (i.e., containing samples from the tasks) and the placebo buffer P (i.e., containing samples from the free image stream) should be matched to the episodic memory size in other comparable methods. The memory budget of the compared methods is not clearly stated. In addition, the "budget" and "non-budget" in the ablation study are also not clearly described (what do you mean by "keep the strict memory budget in each phase"?
- W5: The proposed method seems promoting learning new knowledge. As there is a measure to evaluate the ability to learn new knowledge (e.g., intransigence (Chaudhry et al., 2018)), it is suggested to empirically validate the benefit of the proposed approach by the measure.
- W6: As the method employs reinforcement learning (RL), the computational cost would be considerable. But there is no study about the computational complexity.

**Summary Of The Paper:**

The paper uses data chosen from a free image stream (without labels) to supply knowledge (by the "placebo" samples) for class incremental learning. The empirical results shows that the proposed method exhibits good accuracy in large and high resolution image benchmark datasets such as ImageNet-1k with small memory budget. The paper also presents comprehensive ablation study for each proposed component.

**Summary Of The Review:**

Given the strength and the weaknesses, the paper has decent novelty of using extra free data stream for improving continual learning methods but may be less fairly evaluated (i.e., the evaluation protocol is designed favorably towards the proposed method) with other compared methods.

---

> ### Author Response · Authors · 2021-11-22
> **Our Feedback to Reviewer Dwjy (2/2)**
>
> ### `W4`: ***The combined size of episodic memory and the placebo buffer?***
>
> We are sorry for the confusion.
>
> In the "budget" setting (Row 2 of Table 3), the combined size of episodic memory and the placebo buffer is equal to the episodic memory size in other comparable methods. For example, the episodic memory is 5,000 images on CIFAR-100 (N=5). In the "budget" setting, the total buffer for the unlabeled data $|\mathcal{U}|$ is 500 images, so we randomly remove 500 images from the episodic memory---sacrificing some new class data randomly. The combined size of episodic memory and the placebo buffer is strictly kept equal to 5,000 images. The performance of our PlaceboCIL only reduces 0.16 percentage points on average under the "budget" setting. We think the reason is that the amount of new data is much larger than that of old exemplars, and removing a small batch of new data does not hurt the performance much (Liu et al., 2021b).
>
> We have added these details in Section 5.2 of the revised paper.
>
> &nbsp;&nbsp;
>
> ### `W5`: ***A measure to evaluate the ability to learn new knowledge?***
>
> Thanks for the nice suggestion. We compute the LCA10 (Chaudhry et al., 2018) and show the results in Table R9. We can observe that our PlaceboCIL achieves consistently higher LCA10 than baselines, showing that our PlaceboCIL improves the ability to learn new knowledge.
>
> We have added the results and discussions in Section F (appendix).
>
> **Table R9: LCA10 (Chaudhry et al., 2018) results on CIFAR-100, 5-phase.**
>
> | Method | LwF | iCaRL | LUCIR | LUCIR+AANets |
> | -- | -- | -- | -- | -- |
> | Baseline | 40.02 | 47.79 | 52.06 | 56.44 |
> | PlaceboCIL (ours) | 50.82 | 55.24 | 57.48 | 59.69 |
>
> &nbsp;&nbsp;
>
> ### `W6`: ***The computational cost of RL?***
>
> It is true that using RL requires additional training time. However, its training can be offline and use a different dataset (see Row 9 of Table 3 in the main paper). This actually gave us a promising direction for solving the expensive training of RL in future work: we may pre-learn the policy from synthesized CIL tasks and apply it for selecting placebos in a real CIL task. Once pre-trained, the overhead for applying the policy is little, e.g., 1.3% and 2.2% of the total training time, respectively, on CIFAR-100 and ImageNet (ImageNet-Subset and ImageNet-1k), taking POD+AANets as the baseline.

---

> ### Author Response · Authors · 2021-11-22
> **Our Feedback to Reviewer Dwjy (1/2)**
>
> ### `W1`: ***Out of protocol of CIL?***
>
> Using unlabeled data is an extension of the widely used CIL protocol. It can be regarded as semi-supervised learning in the class-incremental setting. We have 1) new class data coming; 2) old class data being discarded; 3) a stream of unlabeled data available to "help". In our experiments, we aim to show that in this new setting, our proposed algorithm improves multiple baseline methods consistently.
>
> In addition, we would like to highlight the following advantages of our method.
>
> - Our method is generic and can be plugged in different distillation-based methods, including logit-level distillation (e.g., LwF and iCaRL) and feature-level distillation (e.g., LUCIR and PODNet).
>
> - The computational overhead is little when adopting a pre-trained policy. For example, on CIFAR-100, plugging our method to iCaRL with a pre-training policy requires only 1.25% additional training time, and it boosts the average accuracy of "5-phase, 20 exemplars per class" by 4.05 percentage points. We will also run the ablation experiments on ImageNet-Subset. We don't have enough time to finish the experiments during the rebuttal period. We promise to include these results in the final version.
>
> &nbsp;&nbsp;
>
> ### `W2`: ***The choice of source of unlabeled data?***
>
> Thanks for pointing this out while we have actually conducted this research under the assumption that unlabeled images come from a finite image pool, e.g., ImageNet. We think discussing an unlimited pool is not feasible, and using the pool with large domain gaps is not helpful for the target CIL tasks.
>
> Please kindly refer to the ablation results of different free data sources in the second block of Table 3 (main paper). It is worth mentioning when comparing Row 5 with Row 3. We can see removing all overlapping (between the pool and the target CIL dataset) classes almost does not reduce the improvement of using our method (only dropping 0.16 percentage point on average accuracy).
>
> On the other hand, only changing the choices of unlabeled data cannot achieve as good performance as ours. To verify this,  we supplement the results in a new ablative setting: “using random unlabeled data” (from different sources) in Table R8. We can observe that no matter what unlabeled data sources we use, our method consistently performs better than using random unlabeled data.  The reason is that our evaluation function (for these data) is learned to adapt to the change of the ratio between old and new data when the number of phases increases.
>
> We have added these results and discussions in Section G (appendix).
>
> **Table R8: Ablation results (%) on CIFAR-100, *N*=5. "Average" and "last" denote the average accuracy over all phases and the last-phase accuracy, respectively. "All" denotes using all data from ImageNet, and "overlapping" means including samples from the overlapping classes between CIFAR-100 and ImageNet.**
>
> | Method | "LwF, average" | "LwF, last" | "iCaRL, average" | "iCaRL, last" |
> | -- | -- | -- | -- | -- |
> | Baseline | 53.19 | 43.18 | 57.12 | 47.49 |
> | PlaceboCIL (ours, all) | 59.29 | 49.64 | 61.17 | 50.96 |
> | PlaceboCIL (ours, overlapping) | 58.95 | 48.71 | 62.15 | 52.62 |
> | Using random unlabeled data (all) | 58.95 | 48.71 | 62.15 | 52.62 |
> | Using random unlabeled data (overlapping) | 50.80 | 40.94 | 55.70 | 46.47 |
>
> | Method | "LUCIR, average" | "LUCIR, last" | "LUCIR+AANets, average" | "LUCIR+AANets, last" |
> | -- | -- | -- | -- | -- |
> | Baseline | 63.17 | 53.17 | 66.72 | 57.77 |
> | PlaceboCIL (ours, all) | 65.48 | 56.77 | 67.33 | 59.32 |
> | PlaceboCIL (ours, overlapping) | 65.73 | 57.26 | 67.48 | 59.06 |
> | Using random unlabeled data (all) | 64.16 | 55.40 | 66.23 | 57.22 |
> | Using random unlabeled data (overlapping) | 64.23 | 54.68 | 66.58 | 57.08 |
>
> &nbsp;&nbsp;
>
> ### `W3`: ***Using a small memory budget is largely beneficial to the proposed method?***
>
> First, we agree that using a small memory budget is beneficial to the proposed method. We think it could be regarded as an advantage of our method instead of a weakness. Our method can leverage the limited information to select the proper placebo data to compute the distillation loss, improving the model to retain the old knowledge.
>
> Second, our method is still useful when the memory budget is larger. Our method boosts the last-phase accuracy of LwF and POD+AANets by 6.46 and 3.64 percentage points, respectively, in the standard evaluation protocol (20 exemplars per class, CIFAR-100, N=5). Please note this standard evaluation protocol is the same as LUCIR (Hou et al., 2019), PODNet (Douillard et al., 2020), Mnemonics (Liu et al., 2020a), and etc.

---

> ### Comment · Reviewer_Dwjy · 2021-11-28
> **Thank you for the responses**
>
> The reviewer appreciates the authors clarifications on some of my concerns. The reviewer has a following concerns left as follows.
>
> - **Regarding W1-Out of protocol**: by other reviews, there are similar work in the literature (GD+ext and DMC). The authors tried to contrast it but only mention the things that are different from those in terms of the proposed method's perspective. I would like to see how other works address each of the three items differently from the proposed method.
> - **Regarding W1-Computational cost**: if the cost is little, why the experiments take long?
> - **Regarding W3+W4**: As the method exploits additional data sources, the memory budget should be carefully discussed. The reviewer appreciates the clarification. But in Table 2, there is no comparison to GD+ext and DMC. Also the comparison with the DER on the same budget (in the reply to the reviewer **vLx7**) shows the less competitive results than DER despite the large offline pretraining cost.
> - **Regarding W5**: is the LCA the intransigence? Where is the nomenclature (LCA) coming from?
> - **Regarding W6**: how much is the pretraining cost?

---

> > ### Author Response · Authors · 2021-11-29
> > **Additional Feedback (2/2)**
> >
> > ### `Regarding W5`
> >
> > Sorry that we misunderstood your request. Previously, we thought the measure you asked for was the LCA in (Chaudhry et al., 2018a). LCA can measure the ability to learn new knowledge.
> >
> > If we are not wrong this time, the measure should be the intransigence (Chaudhry et al., 2018b). We compute the intransigence (I5) (Chaudhry et al., 2018b) and show the results in Table R10. We can observe that our PlaceboCIL achieves consistently lower I5 than baselines, showing its stronger ability to learn new knowledge. We will run experiments on more baselines and include them in the final revision.
> >
> > *References*
> >
> > *(Chaudhry et al., 2018a) "Efficient lifelong learning with a-gem." arXiv preprint arXiv:1812.00420 (2018).*
> >
> > *(Chaudhry et al., 2018b) "Riemannian walk for incremental learning: Understanding forgetting and intransigence." Proceedings of the European Conference on Computer Vision (ECCV). 2018.*
> >
> > **Table R10: The intransigence (I5) results on CIFAR-100, 5-phase. Lower is better.**
> >
> > | Method | iCaRL | LUCIR |
> > | -- | -- | -- |
> > | Baseline | 0.24 | 0.25 |
> > | PlaceboCIL (ours) | 0.20 | 0.19 |
> >
> > &nbsp;&nbsp;
> >
> > ### `Regarding W6`
> > The cost of pre-training the policy is $O(\mu wGH)$ times higher than the time used for training a target CIL model, where $w$, $G$, and $H$ are the number of RL epochs, the number of pseudo CIL tasks, and the number of iterations on each pseudo CIL task, respectively. $\mu$ is the ratio between the numbers of training epochs in pseudo CIL tasks and in target CIL tasks. For example, on the CIFAR-100 dataset, $w$, $G$, $H$, and $\mu$ are set as 50, 5, 10, and 0.1, respectively. Intuitively, on this dataset, we pre-train the policy for around 650 hours and train the CIL models for around 2.7 hours (using an NVIDIA v100 GPU).

---

> > ### Author Response · Authors · 2021-11-29
> > **Additional Feedback (1/2)**
> >
> > Thanks very much for your new comments. Below is our feedback.
> >
> > &nbsp;&nbsp;
> >
> > ### `Regarding W1-Out of protocol`
> >
> > Indeed we should have clarified more about these two related works.
> > - GD+ext has a new learning objective termed global distillation. In the global distillation, GD+ext (i) trains a teacher specialized for the current phase; (ii) trains a model by distilling the knowledge of the previous model, the teacher learned in (i), and their ensemble; and (iii) fine-tunes the model to avoid overfitting to the data of the current phase. In addition, GD+ext contains a confidence-based sampling method to select unlabeled data. We need to highlight that GD+ext uses double-size memory compared to the baselines---LUCIR (Hou et al., 2019), PODNet (Douillard et al., 2020), and Mnemonics (Liu et al., 2020a).
> >
> > - DMC has a new learning objective function to combine two deep models into a single compact model (which has been validated to improve symmetric knowledge transfer). In addition, it leverages random unlabeled data and doesn't require additional memory.
> >
> > Both GD+ext and DMC introduce new distillation-based losses and use unlabeled data. For unlabeled data, they use either random or heuristic selection methods. In contrast, we focus on the new algorithm of optimizing the selection of unlabeled data. We train the data evaluation functions (using RL) that can adapt to different CIL phases.
> >
> > It is worth noting that GD+ext, DMC and ours are under the similar settings about data (mentioned in our initial feedback): 1) new class data coming; 2) old class data being discarded; 3) a stream of unlabeled data available to "help". Our focus is more on how to select better unlabeled data in a learnable way, while the other two focus more on distillation-based losses. Essentially, our algorithm is orthogonal to using different distillation-based losses. We will try to plug our algorithm into their methods---by replacing their randomly or heuristically selected data with our placebos.
> >
> > &nbsp;&nbsp;
> >
> > ### `Regarding W1-Computational cost`
> >
> > Experiments taking long is because we need to run many models for multiple times to complete the ablation study. Specifically, there are 12 different ablation settings in Table 3, and for each, we need to run four baselines. So we need to run 12 settings $\times$ 4 baselines/setting $\times$ 5 phases/baseline $\times$ 90 epochs/phase, which are around 21,600 epochs in total, on ImageNet-Subset. We promise to include these results in the final revision.
> >
> > &nbsp;&nbsp;
> >
> > ### `Regarding W3+W4`
> >
> > ***Q1: GD+ext and DMC are not in Table 2?***
> >
> > We compared our method with GD+ext and DMC in `General comment 1, Part III: experimental results`. We will add these results to Table 2 in the final version. We need to mention that DMC didn't provide open-source code. We will try to reproduce the results of DMC using the unified benchmark proposed by LUCIR (Hou et al., 2019).
> >
> > ***Q2: The comparison with DER?***
> >
> > We want to highlight two points.
> > - Our method performs especially better when the memory budget is smaller. DER is, however, not applicable to small-budget settings as there is no space to save so many encoders. During the increase of learning phases, DER needs more additional space for saving more encoders and is thus less applicable in real-world applications.
> >
> >
> > - The pre-training of policy takes a high offline cost. However, the trained policy is often transferable. See the discussion in Section 5.2 "Ablation study". It means that we can reuse the policy for related CIL tasks, by which we can save time and make the method more applicable.

---

### Official Review · Reviewer_9D1K · 2021-10-25

**Correctness:** 3
**Technical Novelty And Significance:** 2
**Empirical Novelty And Significance:** 2
**Recommendation:** 3
**Confidence:** 4

**Main Review:**

Positive points:
a) the paper confirms earlier observations that external data can significantly improve results on a wide range of KD-based CL methods.

b) paper is well written, and motivation (fig 1 is nice)

c) ablation shows that RL improves results (though as said above I would have liked more insights and stronger baselines).

The paper has the following weaknesses:

a) Related work: the usage of external data has been studied before in continual learning. I am aware of two relevant works:

[1] Zhang, J., Zhang, J., Ghosh, S., Li, D., Tasci, S., Heck, L., Zhang, H. and Kuo, C.C.J., 2020. Class-incremental learning via deep model consolidation. In Proceedings of the IEEE/CVF Winter Conference on Applications of Computer Vision (pp. 1131-1140).
[2] Lee, K., Lee, K., Shin, J. and Lee, H., 2019. Overcoming catastrophic forgetting with unlabeled data in the wild. In Proceedings of the IEEE/CVF International Conference on Computer Vision (pp. 312-321).

These two methods (and especially [2]) already show the main contribution of the proposed paper, namely that external data can help significantly improve CL. [2] also provides a method to sample from the stream (called 'Confidence calibration for sampling.'). Comparison with these methods is of great importance to evaluate the proposed method. Also, the novelty claim should be adjusted according to these works.

b) I think the RL algorithm was interesting. However, I would have liked to see a more in-depth analysis of its results, comparison also to simpler baselines (choosing only samples with higher confidence for example).

c) It would be nice to also see results for the equal-size split on CIFAR 100 (i.e. 10 tasks of 10 classes). I do not see why the proposed method would perform worse in this setting (because of the RL training ?). Anyways even if underperforming for this setting, it would be insightful to include it.

**Summary Of The Paper:**

The paper proposes to use a separate available data stream (called placebos) to distill information during the continual learning. They show that this is almost as efficient as using ‘old data’, and much more efficient than using ‘new data’ (the normal distillation setting). To balance the different losses they propose a reinforcement learning algorithm that judges the quality of placebos.  They obtain good results on several standard datasets.

**Summary Of The Review:**

Regretfully the authors missed very important related work already showing the potential of external data. This drastically reduces the novelty of the paper. This work also proposes a method to select data from the data stream which could be compared to the proposed RL algorithm. Given that comparison with the most relevant related work is missing, I recommend rejection.

---

> ### Author Response · Authors · 2021-11-22
> **Our Feedback to Reviewer 9D1K**
>
> ### `Q1`: ***The usage of external data has been studied before in continual learning?***
>
> Thanks for pointing out these related papers. Please kindly refer to `General comment 1`.
>
> &nbsp;&nbsp;
>
> ### `Q2`: ***A more in-depth analysis of choosing only samples with higher confidence?***
>
> Thanks for the nice suggestion. We fully agree that it is necessary to take an in-depth analysis of using different strategies to select unlabeled data. Please kindly refer to `General comment 2`.
>
> &nbsp;&nbsp;
>
> ### `Q3`: ***The results for the equal-size split on CIFAR 100?***
>
> We agree that the equal-size split is another interesting setting adopted in IL2M (Belouadah et al., 2019), EEIL (Castro et al. 2020), etc. For the equal-size split setting, we are not possible to run the RL algorithm because there are too few classes to create the pseudo CIL tasks. Instead, we may either use transferred RL policy from another or the heuristic evaluation functions. ($\beta_i=\gamma_i=1$). We conducted the experiments on CIFAR 100, 10-phase (10 classes per phase, 20 exemplars per class).  We can observe that using placebos selected by the heuristic evaluation functions consistently improves the results for three baselines. Using the transferred RL algorithm can further boost performance. The observations are similar to using the "training-from-half" setting in the main paper.
>
> We added these results and discussions in Section E (in the appendix).
>
> **Table R7: The equal-size split results (%) on CIFAR-100, 10-phase (10 classes/phase, 20 exemplars/class). "Average" and "last" denote the average accuracy over all phases and the last-phase accuracy, respectively. "Transferred RL" denotes using the policy learned on ImageNet-Subset to create the evaluation functions. "W/o RL" denotes using the heuristic evaluation functions ($\beta_i=\gamma_i=1$).**
>
> | Method | "LwF, average" | "LwF, last" | "iCaRL, average" | "iCaRL, last" | "LUCIR, average" | "LUCIR, last" |
> | -- | -- | -- | -- | -- |-- | -- |
> | Baseline | 53.85 | 41.00 | 59.70 | 45.29 | 56.71 | 42.78 |
> | PlaceboCIL (ours, transferred RL) | 57.68 | 42.44 | 62.32 | 46.52 | 57.89 | 44.08 |
> | PlaceboCIL (ours, w/o RL) | 56.31 | 41.76 | 61.05 | 46.02 | 57.01 | 43.13 |

---

### Official Review · Reviewer_4jmy · 2021-11-02

**Correctness:** 2
**Technical Novelty And Significance:** 2
**Empirical Novelty And Significance:** 2
**Recommendation:** 6
**Confidence:** 4

**Main Review:**

- The RL-based sampling algorithm looks interesting. The paper is well-written, but some parts could be improved (see suggestions below).

- The first contribution, leveraging "unlabeled placebo data from a free image stream to improve the KD effect in CIL models" has already been studied in a prior work, [Lee et al.]. The second contribution, the RL-based sampling algorithm is not in this prior work, so maybe the authors can take [Lee et al.] as another baseline, though it seems the main performance gain is from the usage of the free image stream rather than the sampling algorithm (by looking at the ablation study in Table 3 & performance gain shown in [Lee et al.]).

- Knowledge distillation can be done in many different forms, so the formulation should be clarified. I assume that the KL divergence loss is used, as 1) they referred to a series of works using it like LwF and iCaRL, and 2) Fig 1 matches the input of the CE loss and KD loss.

- Fig 1(b) is technically incorrect. The KL divergence loss is defined only when two inputs are probability vectors. You should normalize the inputs. If the loss is really computed in this way, the method is technically incorrect.

- Fig 2 (and even the paragraph referencing Fig 2) is not self-explainable, e.g., a description about the way to construct placebos is necessary.

- Could you elaborate more on models used in No. 8 and 9 in Table 3?

[Lee et al.] Overcoming Catastrophic Forgetting with Unlabeled Data in the Wild. In ICCV, 2019.

Just to note, my initial recommendation is under an assumption that the authors address all my concerns well.

## Post rebuttal

As other reviewers agreed, this paper missed some important prior works like [Lee et al.] and [Zhang et al.] at the time of submission. The first contribution "to compute the KD loss using placebo data chosen from a free image stream" is duplicated with them, so the abstract and contribution paragraph in the intro had to be significantly rewritten but they didn't.

However, apart from it, additional comparisons made during the rebuttal period is impressive. Also, I think the second contribution with an RL algorithm is interesting, and the ablation study on it (row 8 and 9 in Table 3) justify its effectiveness properly.


**Summary Of The Paper:**

This paper proposes to sample placebo data from a free image stream and use them for knowledge distillation during class-incremental learning. For effective sampling, an RL algorithm is also proposed. Experimental results show that the proposed method improves state-of-the-art methods.

**Summary Of The Review:**

The idea of sampling data from a free image stream for class-incremental learning is not new, but the RL-based sampling algorithm looks new. The paper is well-written, but some parts could be improved.

---

> ### Author Response · Authors · 2021-11-22
> **Our Feedback to Reviewer 4jmy**
>
> ### `Q1`: ***Similar to [Lee et al.]?***
>
> Thanks for pointing out this related paper.
> Please kindly refer to `General comment 1` and `General comment 2`.
>
> &nbsp;&nbsp;
>
> ### `Q2`: ***The formulation should be clarified?***
>
> Sorry for the lack of detailed explanation about knowledge distillation.
>
> We plug our method into the baselines by keeping their own distillation losses unchanged. (1) If the baselines are LwF (Li \& Hoiem, 2016) and iCaRL (Rebuffi et al., 2017), we use the softmax KL divergence loss. (2) If the baselines are LUCIR (Hou et al., 2019), and LUCIR+AANets (Liu et al., 2021a), we use the cosine embedding loss (Hou et al., 2019). (3) If the baseline is POD+AANets (Liu et al., 2021a), we use pooled outputs distillation loss (Douillard et al., 2020). The baselines in Figure 1 are LwF and iCaRL, so we use the softmax KL divergence loss.
>
> We supplemented these details in Section C "Training Configurations" (in the appendix), and the caption of Figure 1 (in the main paper).
>
> &nbsp;&nbsp;
>
> ### `Q3`: ***The KL divergence loss in Figure 1(b)?***
>
> Sorry for the confusion. We actually normalized the inputs of the KL divergence loss in our experiments. Before computing the KL divergence, we apply a softmax function. For all these details, we follow the official code of LUCIR (Hou et al., 2019), which includes the re-implementation of LwF (Li & Hoiem, 2016) and iCaRL (Rebuffi et al., 2017). We have amended Figure 1 (b) and clarified this in the caption. Please note that using softmax does not solve the problem mentioned in Figure 1 (b). The KD issue still exists. Please kindly refer to the new Figure 1 in the revised paper.
>
> &nbsp;&nbsp;
>
> ### `Q4`: ***Figure 2 is not self-explainable?***
>
> Thanks for the suggestion.
>
> Here we elaborate the way to construct placebos (added to the caption of Figure 2). The steps are as follows: 1) We use the evaluation functions $S_m(u)$ to output the score for each unlabeled sample. 2) We load a batch of unlabeled data $\mathcal{U}$ from the free image stream, and add $K$ placebos with the highest scores for each old class to $\mathcal{P}$. 3) We compute KD loss using the placebos then delete them from $\mathcal{P}$. 4) When we use up the selected placebos, we repeat the selection steps.
>
> &nbsp;&nbsp;
>
>
> ### `Q5`: ***Could you elaborate more on models used in No. 8 and 9 in Table 3?***
>
> These two models are used to test the efficiency (No. 8) and transferability (No. 9) of our RL algorithm. Both models use the same setting and hyperparameters as the baseline (No. 1 and No. 3). In the "w/o RL" setting (No. 8), we use heuristic evaluation functions ($\beta_i$=$\gamma_i$=$1$) instead of performing any RL. Comparing No. 8 with No. 3 shows that using our RL algorithm boosts the model performance In the "w/ Transferred RL" setting (No. 9), we transfer the RL policy learned on another dataset (i.e., the ImageNet-Subset dataset) rather than CIFAR-100, which means on the target CIL task of CIFAR-100, we do not perform any RL. Comparing No. 9 with No. 3, we are happy to see that the RL policy learned is generalizable and can be transferred to the placebo selection on other datasets.

---

> ### Author Response · Authors · 2021-11-29
> **Thanks for your post-rebuttal comments**
>
> Thank you very much for finding our RL algorithm "new" and "interesting", and our additional experiments "impressive". During the rebuttal, we focused too much on conducting additional experiments and comparisons but forgot to revise the texts in the abstract and introduction sections. We really appreciate that you point this out. We will definitely modify these two sections (as well as other parts related to our contribution) in the final paper.

---

### Official Review · Reviewer_vLx7 · 2021-11-03

**Correctness:** 4
**Technical Novelty And Significance:** 3
**Empirical Novelty And Significance:** 3
**Recommendation:** 6
**Confidence:** 4

**Main Review:**

Pros
- The idea of introducing unlabeled placebo data for alleviating catastrophic forgetting seems interesting and novel for class incremental learning. Such extra training data does not require additional data memory.
- This paper is well-written and easy to follow.
- The proposed method achieves strong performances across multiple benchmarks, in particular for high-resolution image classification tasks, with higher memory efficiency for old data.

Concerns
- The related work is a bit lacking and many recent related works are missing:

  [A] DER: Dynamically Expandable Representation for Class Incremental Learning (SOTA Method, better than PODNet+AANet)

  [B] On Learning the Geodesic Path for Incremental Learning (better than or comparable to PODNet+AANet on CIFAR100)

- Some of the previous work has explored other types of unlabeled data in the class incremental learning, which should be discussed:

  [C] Overcoming Catastrophic Forgetting with Unlabeled Data in the Wild (ICCV2019)

  [D] More Classifiers, Less Forgetting: A Generic Multi-classifier Paradigm for Incremental Learning (ECCV2020)

- The effect of RL seems limited. In Table 3, for the method LUCIR+AANets, the performance of RL is only marginally better than w/o RL (<0.5%).  It is unclear if the RL strategy is necessary.

- Some of the experiment details are unclear/missing:
1) The experimental setting of Figure 1 seems unclear: Does the experiment of 'KD on new' keep old data? What are the values of temperature and distillation coefficients and how does the method determine them?
2) The values of some hyperparameters are missing:  What is the batch size for training? How many samples are taken from the free image stream (|\mathcal{U}|) as candidates? In each iteration, how many unlabeled data samples (K) are selected from \mathcal{U} for training?
3) RL comparison: In Table.3, What sampling policy is used in No.8 w/o RL? Is it random sampling?

**Summary Of The Paper:**

This paper proposes to apply distillation loss on unlabeled data to tackle the catastrophic forgetting problem for class incremental learning. In each training iteration, this work first retrieves a candidate pool of unlabeled data from a free image stream and then adopts a learned RL strategy to select high-quality unlabeled data to augment the sampled mini-batch. It conducts extensive experiments and achieves consistent improvements on CIFAR-100, ImageNet-100, and ImageNet-1k datasets.

**Summary Of The Review:**

The proposed CIL strategy, which exploits distillation on unlabeled placebo data, seems novel and effective. However, several SOTA methods are missing in the related work, and the experimental settings and the efficacy of RL need to be clarified.

---

> ### Author Response · Authors · 2021-11-22
> **Our Feedback to Reviewer vLx7 (2/2)**
>
> ### `Q3`: ***The effect of RL seems limited?***
>
> We agree that the effect of RL seems limited in a few settings (e.g., LUCIR+AANets, 20 exemplars/class). The reason may be that the regularization of LUCIR+AANets is too strong, so it isn't easy to influence the effect of regularization with RL. We think RL is clearly helpful in two cases (especially for Case-2) and discuss in the following:
>
> Case-1: RL is more effective on simpler baselines, which are more robust to different real-world applications. In Table R6, we can see that using RL boosts the average accuracy and the last-phase accuracy of LwF by 2.23 and 3.52 percentage points, respectively (CIFAR-100, 5-phase).
>
> Case-2: RL shows more effectiveness when the number of exemplars decreases. In Table R6, we can see that the improvement of the average accuracy increases from 0.42 to 2.07 percentage points when the number of exemplars decreases from 20 to 5.
>
> **Table R6: Ablation results (%) on CIFAR-100, *N*=5. "Average" and "last" denote the average accuracy over all phases and the last-phase accuracy, respectively. The baseline is POD-AANets (Liu et al., 2021a).**
>
> | Method | "20 exemplars/class, average" | "20 exemplars/class, last" | "5 exemplars/class, average" | "5 exemplars/class, last" |
> | -- | -- | -- | -- | -- |
> | Baseline | 66.72 | 57.77 | 60.28 | 48.23 |
> | Ours w/o RL | 66.91 | 58.88 | 62.03 | 50.76 |
> | Ours | 67.33 | 59.32 | 64.10 | 53.41 |
>
> &nbsp;&nbsp;
>
> ### `Q4-1`: ***The experimental setting of Figure 1 seems unclear?***
>
> Sorry for the confusion caused. We modified Figure 1 and its caption (in blue) in the revised paper.
>
> "KD on new" means computing the distillation loss on both new class data and old class exemplars. It is the default setting in the baseline method iCaRL (Rebuffi et al., 2017). The temperature ($T$) and distillation coefficient ($\beta$) are set as 2 and 0.25, respectively. We use the same hyperparameters as in the official code of LUCIR (Hou et al., 2019).
>
> &nbsp;&nbsp;
>
> ### `Q4-2`: ***The values of some hyperparameters are missing?***
>
> Sorry for missing the details of these hyperparameters.
>
> The training batch size is set as 128, following LUCIR (Hou et al., 2019). $|\mathcal{U}|$ and $K$ are set as 500 and 100, respectively. When we use up the 100 selected placebos, we load another $500$ unlabeled samples to the memory and repeat the selection.
>
> &nbsp;&nbsp;
>
> ### `Q4-3`: ***What sampling policy is used in No.8 w/o RL?***
>
> "No.8 w/o RL" means "without the policy learned by RL". We use heuristic evaluation functions to select unlabeled data. To do so, we set $\beta_i$ and $\gamma_i$ (in Eq. 1) as $1$ in the evaluation functions. We supplemented these details in Section 4.1 "building evaluation functions" (the fifth line of Page 5) and Section 5.2 "ablation study".
>
> We have added the above information in Section 5.1 "Network architectures and configurations" in the revised paper.

---

> ### Author Response · Authors · 2021-11-22
> **Our Feedback to Reviewer vLx7 (1/2)**
>
> ### `Q1`: ***Missing related works \[A, B\]?***
>
> Thanks for pointing this out. In the revised paper, we cited and discussed these papers in Section 2. We compared ours with their results in Table 2 (main paper). Below, we summarize the comparison.
>
> ***Part I: comparing with DER [A].***
>
> We agree that DER achieves better performance than POD-AANets. However, DER has to store all learned encoders (learned in all phases) in the memory. DER uses pruning to reduce the size of the final-phase model (in the $N$-th phase), but it does not propose any method to reduce the memory required during the training (in phases $1\sim N-1$). DER thus breaks the protocol of CIL regarding the memory budget. For example, DER requires 2.76M parameters when using ResNet-32 on CIFAR-100, 10-phase (ours needs only 0.46M).  The extra memory is equivalent for storing 2,995 images of CIFAR-100, because (2.76M-0.46M) floats $\times$ 4 bytes/float $\div$ (32$\times$32$\times$3 bytes/images)$\approx$ 2,995 images.  If we add this number of exemplars in our training, we can achieve better performance than DER, especially in the setting of $N=25$ where the models usually suffer more serious forgetting problems.
>
> **Table R4: Average accuracy across all phases (%) on CIFAR-100. The extra memory budget is used to store more exemplars. Our method is POD-AANets w/ PlaceboCIL. We re-implement DER \[A\] under our benchmark protocol using the their official code on GitHub.**
>
> | Method | *N*=5 | *N*=10 | *N*=25 |
> | -- | -- | -- | -- |
> | DER \[A\] | 68.65 | 67.48 | 66.18 |
> | Ours (using the same memory budget as DER)| 68.78 | 68.01 | 67.35 |
>
> ***Part II: comparison with GeoDL [B].***
>
> In Table R5, we compare our method with GeoDL \[B\] (these results have been added to Table 2 in the main paper). We can see that our method achieves consistently better performance on three datasets.
>
> **Table R5: Average accuracy across all phases (%). We store 20 exemplars per class. Our method is POD-AANets w/ PlaceboCIL. The results of GeoDL \[B\] are from their original paper.**
>
> | Method | "CIFAR-100, *N*=5" | "CIFAR-100, *N*=10" | "CIFAR-100, *N*=25" |
> | -- | -- | -- | -- |
> | GeoDL \[B\] | 65.14 | 65.03 | 63.12 |
> | Ours | 67.47 | 65.70 | 64.53 |
>
> | Method | "ImageNet-Subset, *N*=5" | "ImageNet-Subset, *N*=10" | "ImageNet-Subset, *N*=25" |
> | -- | -- | -- | -- |
> | GeoDL \[B\] | 76.63 | 75.40 | 71.43 |
> | Ours | 78.14 | 77.08 | 75.50 |
>
> | Method | "ImageNet-1k, *N*=5" | "ImageNet-1k, *N*=10" | "ImageNet-1k, *N*=25" |
> | -- | -- | -- | -- |
> | GeoDL \[B\] | 65.23 | 64.79 | 60.97 |
> | Ours | 68.61 | 65.69 | 61.76 |
>
> &nbsp;&nbsp;
>
> ### `Q2`: ***Missing related works [C, D]?***
>
> Thanks for pointing out these papers. Please kindly refer to `General comment 1`.

---

### Author Response · Authors · 2021-11-22
**Our Feedback to the General Comments (2/2)**

### `General comment 2`: ***Using different strategies to select unlabeled data.***

Thanks for the nice suggestion. We fully agree it is helpful to take an in-depth analysis of using different strategies for selecting unlabeled data.

Our work, [Lee et al.], and [Zhang et al.] use different strategies: 1) We introduce an RL-based algorithm that learns a policy to produce phase-specific evaluation functions (for unlabeled data). 2) [Lee et al.] selects unlabeled samples based on each sample's confidence score predicted by the previous model. 3) [Zhang et al.] uses unlabeled samples without selection.

In Tables R2 and R3, we show the average and last-phase accuracies of using these three strategies. We can observe that our RL-based strategy achieves better performance compared to others. We think the reason is that our evaluation functions change accordingly when the number of phases increases and the ratio between old and new classes changes.

We have added the results to Table 3 (main paper) and the corresponding discussions to Section 5.2 (results and analyses).

**Table R2: Average accuracy across all phases (%) on CIFAR-100 (*N*=5, 20 exemplars per class).**

| Method | LwF | iCaRL | LUCIR | LUCIR+AANets |
| -- | -- | -- | -- | -- |
| Baseline | 53.19 | 57.12 | 63.17 | 66.72 |
| PlaceboCIL (ours) | 59.29 | 61.17 | 65.48 | 67.33 |
| Using unlabeled samples with higher confidence [Lee et al.] | 56.98 | 60.43 | 63.60 | 66.97 |
| Using random unlabeled samples [Zhang et al.] | 50.99 | 56.27 | 64.16 | 66.23 |

&nbsp;&nbsp;

**Table R3: Last-phase accuracy (%) on CIFAR-100 (*N*=5, 20 exemplars per class).**

| Method | LwF | iCaRL | LUCIR | LUCIR+AANets |
| -- | -- | -- | -- | -- |
| Baseline | 43.18 | 47.49 | 53.17 | 57.77 |
| PlaceboCIL (ours) | 49.64 | 50.96 | 56.77 | 59.32 |
| Using unlabeled samples with higher confidence [Lee et al.] | 46.20 | 49.36 | 55.50 | 58.12 |
| Using random unlabeled samples [Zhang et al.] | 40.22 | 46.64 | 55.40 | 57.22 |

**References**

**[Lee et al.] "Overcoming catastrophic forgetting with unlabeled data in the wild." CVPR 2019.**

**[Liu et al.] "More classifiers, less forgetting: A generic multi-classifier paradigm for incremental learning." ECCV 2020.**

**[Zhang et al.] "Class-incremental learning via deep model consolidation." WACV 2020.**

---

### Author Response · Authors · 2021-11-22
**Our Feedback to the General Comments (1/2)**

We thank the reviewers for finding our paper "novel" (Reviewer Dwjy), with "strong results" (Reviewer vLx7), "well written", and with "clear presentations". Below is our feedback on the general questions. We made the corresponding changes in the revised paper and colorized these changes in deep blue. A preliminary version of the open-source code is available at https://anonymous.4open.science/r/placeboCIL-B973.

&nbsp;&nbsp;

### `General comment 1`: ***Comparing with some related works that use unlabeled data in incremental learning.***

Thanks for pointing out these related papers. We agree that these papers should be discussed in our paper, and their results should be compared with ours. Please kindly refer to our revised paper, where we updated Section 2 (Related Works), Section 5.2 (Results and Analyses), and Table 3. We summarize these updates into three parts in the following.

***Part I: comparing with GD+ext [Lee et al.] and DMC [Zhang et al.]***

[Lee et al.] uses the unlabeled data to compute the distillation loss in CIL. Our work also uses unlabeled data but differs from [Lee et al.] in three aspects:

- Our method is focused on the learnable and transferrable strategies of sampling helpful unlabeled data for CIL. It is generic and can be plugged in different distillation-based frameworks, including both logit distillation (LwF and iCaRL) and feature distillation (LUCIR and PODNet). Below, we also show the empirical results (comparison) to demonstrate our superiority.
- We propose an RL-based algorithm. Its learned policy can adaptively produce phase-specific functions to evaluate the quality of the unlabeled samples. So our selection functions change accordingly when the number of phases increases. [Lee et al.]'s selection strategy remains unchanged when the number of phases increases. [Zhang et al.] uses the unlabeled data without selection. Below, we show our learnable selection functions generally perform better than [Lee et al.] and [Zhang et al.]. Please kindly refer to General comment 2 for the results.
- Our employment of unlabeled data is dynamic and takes little memory out of the total memory budget. We maintain the memory budget strictly the same as the baselines (LUCIR (Hou et al., 2019), PODNet (Douillard et al., 2020), and Mnemonics (Liu et al., 2020a)), by loading fewer samples of new classes. In contrast, [Lee et al.] requires double-size memory compared to the baselines (LUCIR (Hou et al., 2019), PODNet (Douillard et al., 2020), and Mnemonics (Liu et al., 2020a)).

***Part II: comparing with MUC [Liu et al.]***

Our method and [Liu et al.] use unlabeled data for different purposes and in different ways. We use the unlabeled placebo data from a free image stream to improve the positive effect of KD in training CIL models. [Liu et al.] integrates the ensemble of multiple FC classifiers to set regularization-based constraints. It uses the unlabeled data to compute the classifier discrepancy loss by using the predictions of multiple classifiers.

***Part III: experimental results.***

We run the experiments of GD+ext [Lee et al.] and MUC [Liu et al.] using a unified benchmark (proposed in LUCIR (Hou et al., 2019)) based on their official open-source code on GitHub to make the results comparable. We show the results in Table R1. We didn't find the public code of [Zhang et al.]. So we can only compare with their method using the setting in their paper. iCaRL w/ our method achieves 62.3% on CIFAR-100 (10-phase, 10 classes/phase), while the result of DMC [Zhang et al.] is 59.1%.

**Table R1: Average accuracy across all phases (%) on CIFAR-100. Our method is POD-AANets w/ PlaceboCIL. We save 20 exemplars for each class.**

| Method | *N*=5 | *N*=10 | *N*=25 |
| -- | -- | -- | -- |
| GD+ext [Lee et al.] | 63.17 | 58.71 | 51.79 |
| MUC-LwF [Liu et al.] | 59.03 | 53.27 | 49.06 |
| Ours | 67.47 | 65.70 | 64.53 |

---

### Decision · Program_Chairs · 2022-01-20

**Decision:**

Reject

**Comment:**

This paper presents a novel method for class-incremental learning (CIL) with the help of placebo data chosen from a free image stream. Such placebo data are unlabeled and easy to obtain in practice. To adaptively generate phase-specific functions as the accurate estimation of placebos' quality for KD, this paper applies reinforcement learning based on the constraints of the CIL. The effectiveness of the method has been verified on multiple datasets, including ImageNet-1k and ImageNet-Subset with both lower memory and higher accuracy than baselines. The major concern is about the novelty that the unlabeled auxiliary data is not quite new for CIL despite the minor difference in settings and methods. Moreover, the improvements over the baselines are not significant enough, which is a minor concern.